# *Lippia origanoides* and *Thymus vulgaris* Essential Oils Synergize with Ampicillin against Extended-Spectrum Beta-Lactamase-Producing *Escherichia coli*

**DOI:** 10.3390/microorganisms12081702

**Published:** 2024-08-17

**Authors:** Levi Jafet Bastida-Ramírez, Leticia Buendía-González, Euridice Ladisu Mejía-Argueta, Antonio Sandoval-Cabrera, María Magdalena García-Fabila, Sergio Humberto Pavón-Romero, Monica Padua-Ahumada, Jonnathan Guadalupe Santillán-Benítez

**Affiliations:** 1Facultad de Química, Universidad Autónoma del Estado de México, Toluca C.P. 50120, Mexico; levibastida@gmail.com (L.J.B.-R.); qfb.elma@hotmail.com (E.L.M.-A.); mmgafa@yahoo.com.mx (M.M.G.-F.); shpavonr@uaemex.mx (S.H.P.-R.); 2Facultad de Ciencias, Universidad Autónoma del Estado de México, Toluca C.P. 50200, Mexico; lety_sax@yahoo.com.mx; 3Facultad de Medicina, Universidad Autónoma del Estado de México, Toluca C.P. 50180, Mexico; sandoval.mx@gmail.com; 4Laboratorio de Alta Especialidad Hemato-Oncología, Hospital para el Niño, Instituto Materno Infantil del Estado de México, Toluca C.P. 50170, Mexico; 5Centro Médico, ISSEMyM, Toluca C.P. 52170, Mexico; mopa_526@hotmail.com

**Keywords:** antibiotic resistance, antibacterial activity, ESBL, essential oil interactions

## Abstract

(1) Background: Could compounds such as monoterpenes and sesquiterpenes present in essential plant oils inhibit bacterial growth as an alternative to help mitigate bacterial resistance? The purpose of this study is evaluating the in vitro antibacterial effect of *Lippia organoides* EO (LEO) and *Thymus vulgaris* EO (TEO), individually and in combination with ampicillin, against extended-spectrum beta-lactamase (ESBL)-producing *Escherichia coli* strains; (2) Methods: Experimental in vitro design with post-test. The EOs were obtained by hydrodistillation and were analyzed by GC. ESBL-producing *E. coli* strains used were selected from urine cultures and the *bla_CTX-M_* and *bla_TEM_* resistance genes were identified by end point PCR. The disk diffusion method was used for the susceptibility tests. The MICs and MBCs were determined by microdilution test. Finally, the interaction effect was observed by checkerboard assay; (3) Results: A 39.9% decrease in the growth of the strain thymol in TEO and 70.4% in carvacrol in LEO was shown, observing inhibition halos of 32 mm for both EOs. MICs of 632 and 892 μg/mL for LEO and 738 and 940 μg/mL for TEO were determined. Finally, it was observed that, at low doses, there is a synergistic effect between TEO + LEO and EOs + ampicillin; (4) Conclusions: The findings demonstrate that TEO and LEO have an inhibitory effect on ESBL-producing *E. coli*, suggesting that they are candidates for further studies in the formulation of antibiotics to reduce bacterial resistance to traditional antibiotics.

## 1. Introduction

The excessive use of antibiotics has led to the emergence of resistant bacterial strains. In clinical practice, antibiotics are necessary for only approximately 20% of patients seeking treatment, yet they are prescribed in up to 80% of cases. Moreover, in up to 50% of instances, the dosages or duration of treatments are inadequate [1]. The evolution of resistance is a complex process involving high rates of emergence and persistence of resistant bacterial clones. This represents a significant problem, because the multi-resistant strains that develop in one host can be transmitted to others, even crossing species barriers. Most drug-resistant bacteria contain resistance genes in R plasmids that are horizontally transmitted [2,3]. Some of these genes encode enzymes that modify or inactivate the drug, while others prevent its absorption or expel the molecule through modifications in the cell wall [4]. Bacterial resistance is becoming a great threat to global health, each day, the ability to treat common infectious diseases is decreasing. In 2019, 4.94 million deaths from diseases associated with antibiotic-resistant microorganisms were reported [5]. The β-lactams are essential in treating bacterial infections worldwide, accounting for almost 65% of antibiotic use [6]. Nevertheless, numerous microorganisms have already acquired resistance to this class of drugs due to the production of beta-lactamases such as SHV, TEM, and CTX-M. Extended-spectrum beta-lactamase (ESBL)-producing bacteria cause a wide variety of hospital-acquired infections, mainly impacting the bloodstream, wounds, respiratory and urinary tract [6], and these diseases are usually accompanied by high levels of morbidity and mortality. The prognosis for the incidence of infections caused by ESBL bacteria is not very encouraging; in 2020, because of the COVID-19 pandemic, cases of infections caused by this type of microorganism increased by approximately 7% in the community and 32% in hospital infections [7]. Therefore, there is a need to search for complementary treatments to help combat diseases caused by ESBL bacteria. A niche of interest in the search for new antibiotics can be found in the secondary metabolites produced by plants. The essential oils (EOs) of plant species are a mixture of volatile monoterpenes and sesquiterpenes, which have demonstrated antibacterial effects on ESBL strains, e.g., *Eucalyptus globulus* Labill, *Eucalyptus camaldulensis* Dehnh, *Mentha pulegium* L., *Trachyspermum ammi* L. and *Thymus capitatus* L. [8]. However, there are few studies regarding the antibacterial efficacy of EOs from *Lippia organoids* and *Thymus vulgaris* and their synergism with antibiotics. The EOs of these plant species contain large amounts of thymol and its isomer carvacrol, which can easily cross bacterial cell membranes, causing alterations in their integrity. On the other hand, it has been reported that a decrease in dose through a synergistic interaction of two EOs can favor their use in in vivo tests, due to banking on a dose an appropriate amount for pharmaceutical development [9]. Therefore, this work aimed to evaluate, in vitro, the antibacterial effect of two EOs, alone and in combination with ampicillin, against ESBL-producing *Escherichia coli* strains.

## 2. Materials and Methods

### 2.1. Plant Materials and Essential Oil Extraction (EO)

The method MGA-FH 0090 described in the Pharmacopoeia of the United Mexican States was followed by EO extraction [6]. The dry biomass of *Thymus vulgaris* was acquired from REDSA, SA de CV. (Lot TSU001040422), while *Lippia origanoides* was obtained from ORE S.DE R.L. MI. (Certificate EUK77WCC). Both species were collected in Mexican territory. Briefly, hydrodistillation was performed with a Clevenger trap for EO less dense than water. Dried flowers and leaves of each plant species were ground and sieved. The sample with distilled water was allowed to stand for one hour at room temperature and hydrodistillation was carried out. The EOs were dried with anhydrous sodium sulfate and stored in amber glass vials at 4 °C until use. For MIC-MBC and interactions test, the EOs were dissolved and emulsified with 0.5% and 20% Tween 80, respectively.

### 2.2. Gas Chromatography (GC) Analysis

Analysis of Eos’ chemical compounds was carried out using a Perkin Elmer Instruments AutoSystem XL chromatograph with a VF-5 ms factor FOUR^TM^ column (30 m × 0.25 mm, 0.25 µm; Varian). EOs were diluted 1:1000 with hexane, and 1 µL of each sample was injected with the Split open. The separation conditions were as follows: using hydrogen as a carrier gas, an initial temperature of 40 °C was established, followed by two heating ramps, the first one of 5 °C/min to 100 °C, and the second one of 5 °C/min to 150 °C and maintained for 5 min. The total time was 32 min. Thymol and carvacrol were identified with the association of the retention times of reference standards (PHR1134/42632 Supelco^®^, Darmstadt, Germany). Quantification of the compounds was performed from calibration curves constructed with dilutions of the reference standards with different concentrations (62.5–1000 mg/L), and the content in percentage was determined by the internal normalization procedure with R2 values of 0.9958 for thymol and 0.9857 for carvacrol.

### 2.3. Bacterial Strains

Twenty ESBL-producing *E. coli* strains, isolated and identified at the ISSEMyM Toluca Medical Center, were used. Isolation was performed from urine cultures and body secretions; identification was performed using VITEK^®^ 2 Compact automated equipment (BioMérieux, Marcy-l’Étoile, France). Strains were preserved with a 1:1 glycerol ratio with an overnight culture in nutrient broth at −18 °C. The strains were activated on trypticase soy agar for subsequent assays and incubated at 37 °C for 18–24 h before testing.

### 2.4. Antibacterial Activity by the Disk Diffusion Method

Antibacterial activity was evaluated in triplicate by the disk diffusion method, applying some modifications of the methodology established in CLSI M100 [10]. Colonies isolated from the selected strains were picked and plated in sterile saline solution until the McFarland standard (1.5 × 10^8^ CFU/mL) was reached. The inoculum was spread on mats in duplicate on 100 × 15 mm Müeller–Hinton agar plates (BD Bioxon, Ciudad de México, México). Each plate was placed with three sterile disks impregnated with 10 μL of the EOs. Discs with 30 μg of amikacin were used as a positive control; the negative control was discs with sterile saline water. The plates were incubated at 37 °C from 16–18 h; after this time, the inhibition zones were measured considering the diameter of the halos. The control strain was *E. coli* ATCC 25922.

### 2.5. Minimal Inhibitory Concentration (MIC) and Minimal Bactericidal Concentration (MBC) of Essential Oils

The broth microdilution technique of CLSI M07 [10] was used with some modifications. Concisely, the inoculum was prepared as described above, and dilutions were made to obtain 1.5 × 10^5^ CFU/mL. In 96-well plates, 90 µL of cation-adjusted Mueller–Hinton broth (CAMHB) was placed in the first well of each row, 100 µL of each EO was placed in the first well of each row, and serial dilutions were performed, obtaining concentrations of 23.625–46 µL/mL of TEO and 22.830–44 µL/mL for LEO. Each well received 10 µL of inoculum, leaving a final volume of 100 µL in each well. Amikacin (16 µL/mL) was used as a positive control, and a well without antibacterial agents was used as a negative control. Additionally, a vehicle control with Tween 80 was used. The plate was incubated for 18 ± 2 h. MIC was the lowest concentration of the EO that inhibited bacterial growth to the naked eye. The MBC was determined by reseeding 50 µL from the wells in which no growth was observed on agar plates after 24 h.

### 2.6. Interaction Assay

A checkerboard assay was performed according to the methodology of Bellio et al., 2021 [11]. A total of 100 μL of CAMHB 2X medium was placed in a 96-well microplate. For dispensing the EOs, 100 μL of LEO microemulsion (SA) from A1–A11 and 100 μL of 2X LEO (SA 2X) were added in A12; in each column, a dilution was made with 100 μL of A–G removing the excess volume. The TEO microemulsion (SB) was placed in column 12 of A–H, and a second 12-2 dilution was made in all rows. A total of 100 μL of bacterial inoculum with a 5 × 10^6^ CFU/mL concentration was added to each well, and H1 was used as a growth control. Microplates were incubated at 35 ± 2 °C for 18 ± 2 h. Subsequently, OD was measured at 600 nm in a VICTOR^®^ Nivo (Waltham, MA, USA) plate reader. The same procedure was followed for the interaction between EOs and ampicillin, considering TEO and LEO as SA/SA 2X separately and the antibiotic as SB. The concentrations tested were 17,175–14 µg/mL for TEO, 13,698–13 µg/mL for LEO, and 1–1024 µg/mL for ampicillin.

Percent growth was determined by:(1)$$\frac{OD\;combination\;SA-SB-OD\;no\;growth}{OD\;no\;combination\;SA-SB-OD\;no\;growth}\times 100.$$

The OD no growth was obtained from the plate without inoculum. The MICs for each EO combination was the concentration of EO that reduced growth by 80% compared with that of organisms grown without EO [10].

### 2.7. Loewe Additivity Model

To describe the interactions between the EOs, the Loewe model as explained by Segatore et al., 2012 [12] was used. The mathematical model states that an agent cannot interact with itself, so the interaction index *I* is defined as the sum of the ratios of the concentration of each drug in the mixture that produces a certain percentage of inhibition to the concentration of each drug that produces the same percentage of inhibition when used separately. Thus, when *I* > 1, it indicates an antagonistic interaction, and when *I* < 1, it indicates a synergistic relationship. *I* can be adapted to calculate the Fractional Inhibitory Concentration Index (FICI), which expresses the effect of the combination of antibacterial agents. Therefore, when a synergistic relationship exists, the FICI will be the lowest ΣFIC, while for antagonistic relationships, the highest ΣFIC will be considered. Thus, when FICI ≤ 0.5, there is a synergistic correlation between the antimicrobial agents; when FICI > 4, the correlation is antagonistic; and when 0.5 < FICI ≤ 4, there is indifference [12,13].

### 2.8. Detection of Antibiotic Resistance Genes β-Lactamics

Plasmid DNA was obtained by alkaline lysis extraction according to the method of Mejía-Argueta et al., 2020 [14]. Oligonucleotides were designed in Primer3Plus: *bla_TEM_* (F: GAT AAC ACT GCG GCC AAC TT and R: TTG CCG GGA AGC TAG AGT AA), *bla_SHV_* (F: CTT TCC CAT GAT GAT GAG CAC CT and R: CGC TGT TAT CGC TCA TGG TA) and *bla_CTX-M_* (F: TTG TTA GGA AGT GTG CCG CT and R: AGG TGA AGT GGT ATC ACG CG).

The endpoint PCR conditions for the detection of resistance genes were an initial denaturation at 94 °C for 5 min followed by 30 cycles of amplification with denaturation at 94 °C for 30 s, hybridization at 55 °C for 40 s, and extension at 72 °C for 1 min, followed by a final extension at 72 °C for 5 min.

### 2.9. Statistical Analysis

Experiments and determinations were performed in duplicate. For the analysis of the MICs, a Kruskal–Wallis test was used, and to compare the resistance genes, the Mann–Whitney U test was used to compare medians between the EOs. IBM SPSS 29.0.2.2, 2023 statistical software was used, with statistical significance at *p* ≤ 0.05.

## 3. Results and Discussion

### 3.1. Carvacrol and Thymol Are the Major Components in LEO and TEO, Respectively

The yield of essential oil extraction was 1.2% ± 0.08 for TEO, while for LEO it was 3.6% ± 0.02. Yields above 2% are considered high. It was previously reported that the European variety of *O. vulgare* can contain essential oils in a range from 0.03% to 4.6% [15]. Therefore, *L. origanoides* appears to be a better alternative than *T. vulgaris* for obtaining aromatic compounds. After GC determined the content of carvacrol and thymol in TEO and LEO. It was found that the major component in LEO was carvacrol (70.4%) with a RT of 24.8 (Table 1). This content is like that reported by Calvo et al., 2014 [16], who found 72.20–83.49% concentrations in the EO of *L. graveolens*. Moreover, for TEO, the main component was thymol (39.4%) (Table 1). When thymol is the significant component, it can range from 23–60% [17]. It has been demonstrated that thymol and carvacrol have antibacterial properties, making it advantageous for them to be the major components of EOs, as their high concentration can enhance their antimicrobial effectiveness, even when used in low volumes. Although the presence of thymol and carvacrol was confirmed in the essential oils of *T. vulgaris* and *L. organoides*, the concentration depends on geobotanical conditions, the vegetative cycle, and the extraction method [18].

The LEO chromatogram shows that the main peak is from carvacrol, whereas the TEO chromatogram showed thymol as the main component with a RT of 24.4; additionally, we found a peak with an area reflecting a percentage of 24.6% with a RT of 12.2 min (Figure 1). However, the compound’s identity could not be confirmed due to the lack of a reference standard. Based on the literature and considering the retention times, it is plausible that the compound in question could be p-cymene [17], a precursor of carvacrol. In general, p-cymene has been attributed as having analgesic, anti-inflammatory, and antibacterial properties [19]; although, since it lacks a hydroxyl group, it is not as effective as thymol or carvacrol. Therefore, it is suggested that the antibacterial properties outlined in the following sections are attributable to the presence of thymol as the predominant component of TEO and carvacrol as the predominant component of LEO.

### 3.2. Bla Genes Are, in the Tested Clinical Isolates, Resistant to β-Lactam Antibiotics

Due to resistance to beta-lactams, ESBL-producing *E. coli* complicates treatment for patients, requiring alternative or combined medications. This growing resistance has increased the incidence of difficult-to-treat urinary tract infections, which can lead to more severe complications. Eighty-five percent of the strains used in this study were isolated from urine cultures, and the remaining 15% were isolated from other secretions. The strains were resistant to all beta-lactams tested (ampicillin, cephalothin, cefotaxime, ceftazidime, ceftriaxone, and cefepime); however, when the ampicillin/sulbactam combination was tested, four were sensitive, and four more were sensitive with increased exposure to this combination. Similarly, a study published by Mejía-Argueta et al., 2022 [20] mentioned that out of 135 ESBL-producing *E. coli* strains, 10.4% were sensitive to the ampicillin/sulbactam combination and 35.9% were sensitive to increased exposure when tested with the same combination.

Resistance genes present in ESBL-producing *E. coli* strains were identified from plasmid DNA. Of the strains, 65% were positive for *bla_TEM_*, and 15% were positive for *bla_CTX-M_*. Of the 20 strains, two were amplified for both *bla_TEM_* and *bla_CTX-M_* (Figure 2). The positive control and the resistance genes found were compared with the genes of a previously identified multidrug-resistant strain. The *bla_SHV_* gene was also searched; however, none of the strains were amplified for this gene. Furthermore, previous studies based on clinical isolates in the Mexican population have found that the most prevalent resistance gene for ESBL-producing *E. coli* is *bla_CTX-M_*, followed by *bla_TEM_* [21], and *bla_SHV_* has also been reported to be the least prevalent gene [22]. Although reports point to *bla_CTX-M_* as the most frequent gene, in this study, the most frequent gene was *bla_TEM_*. It should be noted that the prevalence of genes may vary depending on the geographic area [23]. The fact that the isolated strains were resistant to the tested beta-lactams indicates a high prevalence of such strains in hospital environments, which may complicate the prognosis for hospitalized patients, particularly considering the elevated infection rates observed following the COVID-19 pandemic [7].

### 3.3. Inhibitory Activity of TEO and LEO on ESBL Strains

EOs rich in volatile monoterpenes such as thymol and carvacrol typically exhibit antibacterial activity due to the disruption of the lipid fraction of the plasma membrane, resulting in the leakage of intracellular materials [24]. EOs was evaluated in triplicate. For LEO, inhibition halos were found with a mean of 32 mm (SD ± 3.3). The inhibition halos for TEO also had a mean of 32 mm (SD ± 4.8); after comparison of means, no statistically significant differences were found at a 95% confidence level, suggesting that the effect of EOs was the same for the disk diffusion test. The inhibition halo of the EOs surpassed in diameter the halo of β-lactams such as ampicillin and cefepime, which must be ≥17 mm and ≥25 mm, respectively, to be considered effective against *E. coli* [10]. This may indicate that EOs rich in thymol and carvacrol are more effective at inhibiting the growth of ESBL-producing *E. coli* compared to beta-lactams, which have become obsolete for these types of strains. This suggests that these extracts could be a valuable option for treating infections where traditional antibiotics have limited efficacy.

The data obtained for LEO can be compared with studies performed with *O. vulgare* EO (OEO) because carvacrol is also usually its main component [25]; inhibition halos of 23 mm have been reported for ATCC 8739 *E. coli* strains [26]. Likewise, TEO was tested with the thymol chemotype, and 21–30 mm halos were observed for ESBL-positive *E. coli* [27]. The data in the literature are comparable to those in this study, which indicate that EOs rich in thymol or carvacrol have antibacterial properties against *E. coli* ATCC and ESBL-producing *E. coli* strains.

Even so, an unexpected outcome was the presence of a growth gradient known as swarming (Figure 3). However, the European Committee on Antimicrobial Susceptibility Testing recommends considering the outer halo [28].

We only worked with 2 discs with EO since when placing more than two discs, the growth decreased throughout the plate; when six discs were used, the strains were inhibited entirely. This could be due to the volatile nature of the essential oils, since being confined to the culture plate, a microenvironment could be created in which the volatized molecules inhibit growth in general. Mollea et al., 2022 [29], also observed this phenomenon and showed that even when OEO and TEO had no contact with the medium, they inhibited the growth of *Staphylococcus epidermidis*. This supports the idea that using LEO and TEO in the disk diffusion method with ESBL-producing *E. coli* creates large inhibition halos and a generalized growth decrease. Furthermore, the antimicrobial effectiveness combined with the volatility of essential oils opens the possibility of exploring new methods of application, such as in aerosols or diffusers, which would allow their use in a variety of contexts and for different types of infections.

### 3.4. ESBL-Producing E. coli Could Be Susceptible to T. vulgaris L. and L. origanoides EOs in Drug Interaction and In Vitro Susceptibility Tests

In this study, MIC and MBC of TEO and LEO were determined, finding low concentrations of MICs for strains by gene groups; ATCC 25922 strain was used as a control (Table 2). This finding is important because it has been suggested that the in vitro concentrations must not be high to enter in vivo assays. After all, although they are not considered dangerous, EOs can be toxic due to their easy overdosage; the MIC would have to be below 1 mg/mL to extrapolate from in vitro tests to in vivo assays successfully [25]. It was also found that there are no significant differences with *p* > 0.05 in the effect of EOs and strains, so it can be assumed that the effect of each EO is the same regardless of the resistance gene, even if it has not been identified. This may imply that, regardless of the resistance gene expressed by *E. coli*, essential oils maintain their antibacterial effectiveness against these strains.

A comparison of the microdilution plate assay with the 20 strains proved significant differences between LEO and TEO with *p* < 0.05, with LEO being the oil with lower MICs. Previously, in liquid cultures, carvacrol, over thymol, had a better effect *against P. aeruginosa* biofilms [30]. This phenomenon may be due to the isopropyl group at position 2 generating steric hindrance around the phenol, which in turn would obstruct the hydroxyl group [31], which is related to the antibacterial activity of thymol. Furthermore, in this study, LEO has a higher concentration of the interested molecule than TEO. Similarly, the yield of LEO is higher than of TEO, suggesting that LEO may be a better alternative as an antimicrobial agent.

Similarly, the MICs of both EOs were reduced because of their combination with each other or in combination with ampicillin. However, there was no decrease in the MIC of ampicillin below 32 µg/mL, a concentration at which *E. coli* is considered resistant.

In the Bliss synergism and antagonism model, it was observed that at low concentrations, ampicillin and EOs have synergistic interactions, which become antagonistic interactions as the concentration increases (Figure 4A). Considering the % control, the positive points indicate synergy and the hostile ones, antagonism; likewise, in the Bliss synergy and antagonism model the interactions <100% were considered weak, between 100% and 200% were considered moderate, and >200% strong [13].

In the TEO × Amp combination, an average FICI value of 0.333 was found, while for the LEO × Amp combination, the value was 0.237, and for the LEO × TEO combination, it was 0.27. These values are interpreted as synergism [13]. It is worth noting that Figure 4 shows synergistic interactions at lower concentrations of the combinations (LEO × TEO, LEO × Amp, TEO × Amp). This suggests that the synergy is more pronounced at specific concentrations that may not be fully reflected in the FICI calculation.

Previous studies have shown that thymol and carvacrol molecules synergize by decreasing concentrations to inhibit bacterial growth [32]. This study found that this synergistic effect is maintained even when the combination is between EOs and not between isolated compounds. Likewise, a synergistic effect has been reported between EOs of *E. globulus* and *M. alternifolia* in combination with vancomycin, cefotaxime, and oxacillin [27,32], and the combination of EOs of *O. vulgare* with levofloxacin and doxycycline [33].

The synergy observed with EOs in combination with ampicillin may be due to the alteration of the permeability of the external membrane caused by molecules such as thymol and carvacrol, which would facilitate the arrival of the antibiotic to the cell wall [34], improving the action of ampicillin despite the presence of beta-lactamases.

The high concentrations of the EOs may saturate the medium, hindering the arrival of the ligands to their molecular targets. This phenomenon is also related to the use of Tween 80 as a nonionic surfactant. Although it has been observed that this can maintain stable nanoemulsions of EOs, it is not ruled out that these may lose stability by forming aggregates, thus decreasing the contact of the bioactive molecules with the bacterial cells, making the combined effect of LEO and TEO, or ampicillin with the EOs, less than the sum of the effects of each one. It was previously observed that nanoemulsions of AEs with Tween 80 can lose stability due to storage time or aggregation of ions or other components in the medium [35]. Regardless, the greatest effect was observed at low concentrations of the tested essential oils. This can be advantageous, as using lower concentrations could reduce the possibility of toxicity in in vivo studies. Additionally, only small volumes would be needed, which is beneficial given that some essential oils have low yields. This characteristic not only optimizes the efficiency of the studies but can also contribute to improved sustainability and economy in the production and application of EOs.

## 4. Conclusions

We have demonstrated that TEO and LEO have an antibacterial effect against ESBL-producing *E. coli* strains, inhibiting plate growth and exceeding the diameter of common beta-lactam inhibition halos. Consequently, there is a general decrease in growth throughout the medium, which opens up the possibility of investigating the antibacterial effect focused on the volatile properties of EOs. Since thymol and carvacrol are molecules that, upon contact with bacterial cell membranes, initiate a process of bacterial lysis, our results also suggest that when the EOs used in this work are combined in the liquid medium, they may be interacting synergistically, thus decreasing the MIC. Furthermore, the combination of low doses of the EOs and EOs + ampicillin resulted in synergistic interactions against strains with *bla_TEM_*, *bla_CTX-M_*, and *bla_TEM_* + *bla_CTX-M_* genes. The reduction in the volume needed to achieve therapeutic effects can also facilitate their formulation and administration in various forms, thus expanding their applicability in different clinical and therapeutic contexts. In future work, it will be crucial to delve deeper into the mechanisms of action of these combinations to better understand how they interact with bacteria. Additionally, a broader analysis of other *E. coli* strains of the same species should be conducted to evaluate whether the observed effects are strain dependent. This expanded research could lay the groundwork for using traditional medicine derivatives in the preclinical development of new antibiotics or in creating new strategies to extend the use of beta-lactams against resistant strains, providing a more detailed understanding and enabling the development of more effective and specific treatments against antibiotic-resistant bacterial infections.

## Figures and Tables

**Figure 1 microorganisms-12-01702-f001:**
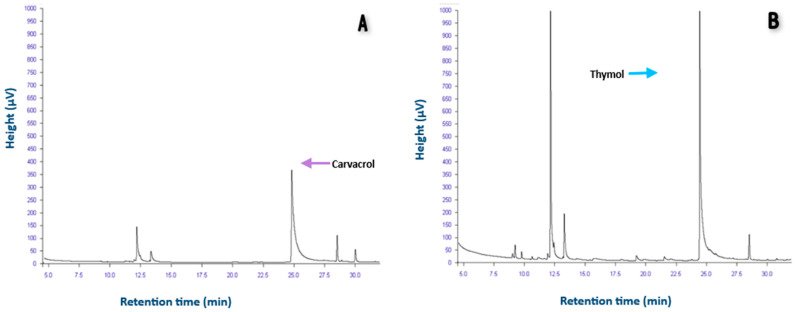
Major components of the EOs. (**A**) A chromatogram of LEO; the primary component, carvacrol, showed a retention time of 24.8 min. (**B**) A chromatogram of TEO; the significant component thymol, recorded a retention time of 24.4 min.

**Figure 2 microorganisms-12-01702-f002:**
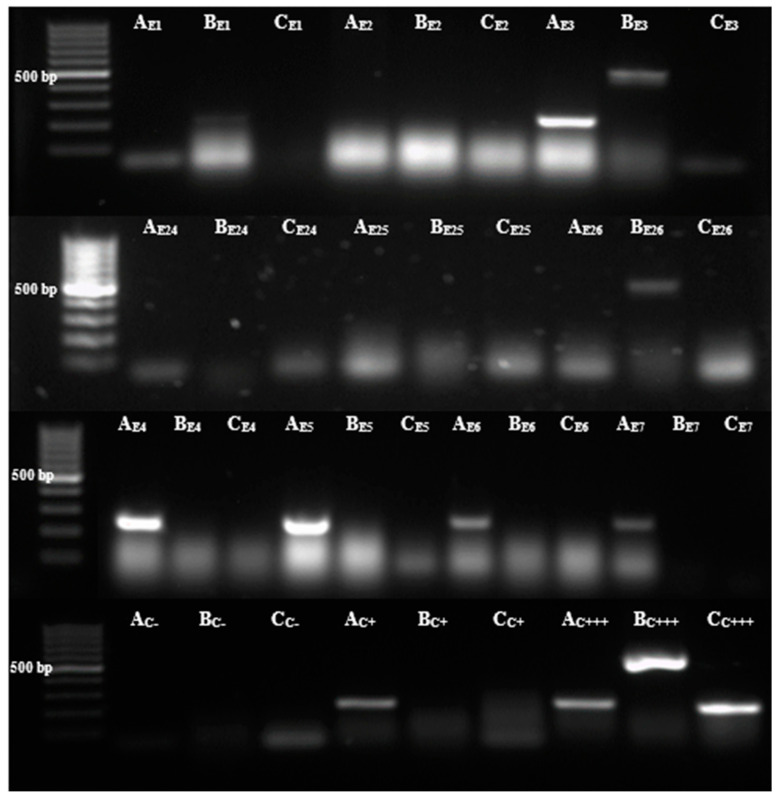
Resistance genes were found in ESBL-producing *E. coli bla_TEM_* (A), *bla_CTX-M_* (B), *bla_SHV_* (C), negative control (C− and positive control (C+++). Strain E3 and E16 amplified for *bla_TEM_* and *bla_CTX-M_*. Thirteen strains, including E4, E5, and E6, were amplified for *bla_TEM_*, and E26 was only amplified for *bla_CTX-M_*.

**Figure 3 microorganisms-12-01702-f003:**
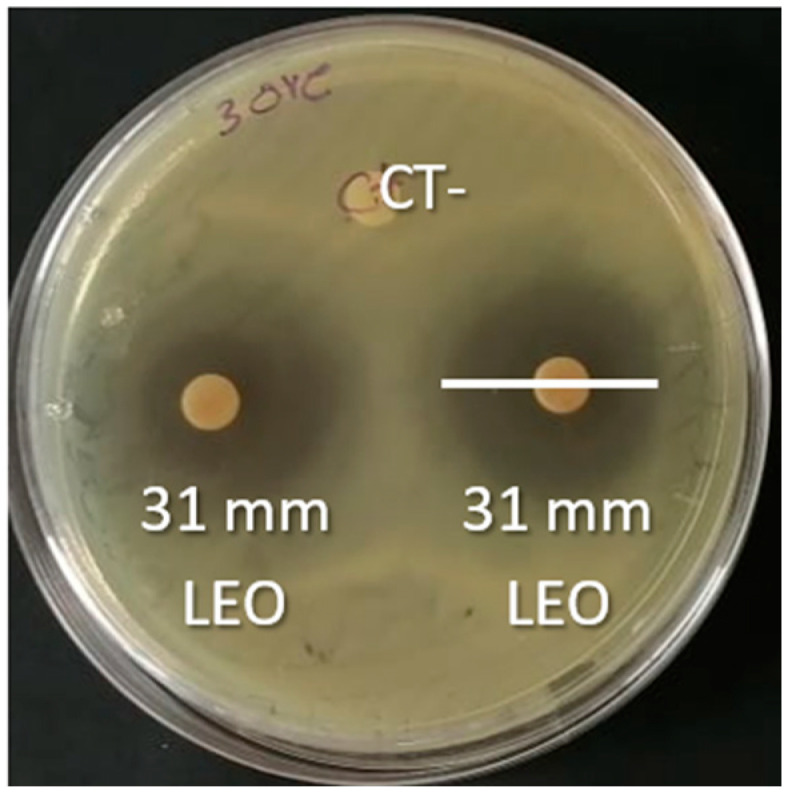
Inhibition halo with swarming corresponding to LEO and a disk without halo corresponding to the negative control.

**Figure 4 microorganisms-12-01702-f004:**
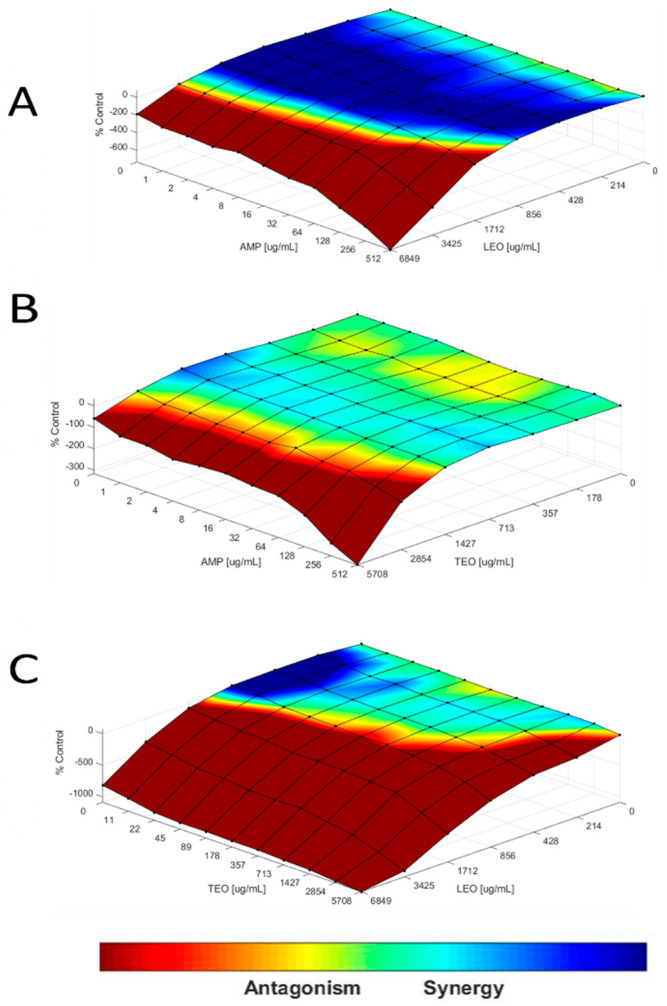
Bliss model of synergism and antagonism. (**A**) LEO vs. Amp for *bla_TEM_*, (**B**) TEO vs. Amp for *bla_TEM_* + *bla_CTX-M_* y, (**C**) LEO vs. TEO for *bla_CTX-M_*.

**Table 1 microorganisms-12-01702-t001:** Thymol and carvacrol content in LEO and TEO from leaves and flowers, respectively.

Compound	LEO ^1^ (mg/L)	TEO ^1^ (mg/L)	LEO (%)	TEO (%)
**Thymol**	62.9 ± 0.2	828.4 ± 0.4	0.005	39.9
**Carvacrol**	612.2 ± 0.2	112.4 ± 0.6	70.4	4.6
**p-cymene**	--	--	24.6	12.2

^1^ Concentrations present in the 1:1000 dilution with hexane for each EO. Data represent the mean ± SD of two replicates. (--)—data not calculated.

**Table 2 microorganisms-12-01702-t002:** MIC and MBC of ESBL-producing *E. coli* with *blaCTX-M* and *blaTEM* genes for LEO and TEO.

Gene	LEO		TEO	
	MIC µg/mL	MBC µg/mL	MIC µg/mL	MBC µg/mL
** *bla_TEM_* **	632	1265	940	1275
***bla_TEM_*** **+ *bla_CTX-M_***	892	1784	923	2215
** *bla_CTX-M_* **	713	1427	738	1477
**Unidentified**	815	1733	844	1582
**ATCC 25922**	357	713	369	738

## Data Availability

The presented data are original and owned by the author. The generation of images is of own authorship.

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
