# Peer review of "Lippia origanoides and Thymus vulgaris Essential Oils Synergize with Ampicillin against Extended-Spectrum Beta-Lactamase-Producing Escherichia coli"

_microorganisms, 2024, doi:10.3390/microorganisms12081702_

Round 1

Reviewer 1 Report

Comments and Suggestions for Authors

The comments and suggestions for corrections are attached.

Author Response

Thank you for your time and effort in reviewing this manuscript. Please find below the detailed responses, with the corresponding revisions and corrections highlighted in the re-submitted documents.

Comments 1: Introduction, lines 33-43: Regarding bacterial resistance, include more information

Thank you for pointing this out. We agree with this comment. Therefore, we have included mechanisms of resistance and inadequate use of antibiotics.

Comments 2: Introduction, lines 51-52: Cite the scientific names of the following species: Eucalyptus globulus, Eucalyptus camaldulensis, Mentha pulegium, Trachyspermum ammi.

We have corrected the scientific names of the mentioned species.

Comments 3: Materials and Methods, 2.1 Plant materials and essential oil extraction (EO)

Thank you for your comment. We have included the yields of the oil extractions; however, we omitted the identification registration number of the plants and the geographical coordinates of the collection site because the dry biomass was acquired from a certified supplier.

Comments 4: Materials and Methods, 2.1 Strains

Thank you for raising this important concern. We acknowledge that the potential for Mycoplasma contamination in trypticase soy agar is a valid issue. In our study, we did not specifically mention the quality control measures for Mycoplasma contamination in the initial manuscript. However, it is important to clarify that our laboratory follows standard protocols to minimize and monitor contamination risks. This includes routine checks and validation procedures to ensure the integrity of our cultures.

Comments 5: Materials and Methods, 2.5 Minimal inhibitory concentration (MIC) and minimal bactericidal concentration (MBC) of essential oils

We have updated the manuscript to include these details and clarify the use of the vehicle control. Thank you for bringing this to our attention.

In our study, we utilized Tween 80 to dilute the essential oils. Additionally, to account for any effects of the vehicle itself, we included a "vehicle control" in our experimental setup.

Comments 6: Materials and Methods, 2.8 Statistical analysis, lines 151-154:

Thank you for your comment. We have updated the manuscript to specify the supplier, version, and year of the software used in our analysis. Additionally, we have included the adopted "p" value for statistical significance.

Comments 7: Results, 3.1 Carvacrol and thymol are the major components in LEO and TEO respectively

Thank you for your suggestion. We have included additional information about the compounds. Unfortunately, we cannot include all the compounds because only gas chromatography was used in the study.

Comments 8: Results, Figure 2

Thank you for your observation. The suggested changes have been made.

Comments 9: Results, Figure 3

 Thank you for your feedback. The updated figures and descriptions in the revised manuscript should now provide a clear differentiation between these components.

Comments 10: Discussion, Figure 4

Thank you for your suggestion. We have added a legend to the graph in the revised manuscript to indicate the meaning of the different colors. This should help clarify the data representation and make the graph more informative.

Comments 11: Discussion

 Thank you for your valuable feedback. We have enhanced the Discussion section in the revised manuscript by relating our findings to other relevant studies.

Comments 12: References

Thank you for your comment. We have standardized all the references according to the journal's format and ensured uniformity throughout the manuscript.

Reviewer 2 Report

Comments and Suggestions for Authors

The present study investigated the synergistic effect of two essential oils (Lippia organoides EO, LEO) and Thymus vulgaris EO (TEO), individually and in combination with ampicillin, against extended-spectrum beta-lactamase (ESBL)-producing Escherichia coli strains. The manuscript is scientifically written in general. However, a severe flaw that I found is that the antibacterial test was too simple (the disk diffusion test and microdilution test). Despite the multi-drug resistance microorganisms (MDR) being a worldwide concern, using essential oils in such general and superficial methods is inadequate for the journal. One of the most critical things for antibacterial analysis, beyond the halo inhibition and synergistic studies, is also the mechanism of action, cytotoxicity of the essential oils, and novelty of the treatment proposed. I do not see the novelty in this work. Unfortunately, I do not suggest this manuscript for publication unless the mechanism of action of the essential oils is studied. 

Comments on the Quality of English Language

The English language is acceptable. Just minor adjustments are needed.

Author Response

Thank you for your detailed feedback. We appreciate your insights regarding the antibacterial testing methods used in our study.

The primary objective of our study was to investigate the synergistic effects of Lippia organoides essential oil (LEO) and Thymus vulgaris essential oil (TEO), individually and in combination with ampicillin, against ESBL-producing Escherichia coli strains. We acknowledge that understanding the mechanism of action and cytotoxicity of the essential oils would provide additional depth to our findings. However, identifying the specific mechanisms of action was beyond the scope of the current study.

Our focus was on establishing the potential effects using disk diffusion and microdilution tests. These methods were chosen to provide a preliminary but essential insight into the potential efficacy of the essential oils and their combinations with antibiotics against MDR strains.

We agree that further research is needed to elucidate the mechanisms of action and assess the cytotoxicity of these essential oils. We believe that our findings lay the groundwork for such future studies, which could further validate the therapeutic potential of these essential oils.

Reviewer 3 Report

Comments and Suggestions for Authors

The manuscript 3112360 entitled “ Lippia origanoides and Thymus vulgaris essential oils synergize  with ampicillin against extended-spectrum beta-lactamase-producing Escherichia colireports the chemical composition of the essential oils of L. origanoides and T. vulgaris , and their action against clinical E. coli isolates that carry beta-lactamase genes.

The manuscript shows several drawbacks that require major revision, namely part of the results is shown in the discussion section; MIC and MBC values and the interaction with the antibiotic. For interaction the FIC index is mandatory.

Moreover, the English language requires a deep revision.

Comments on the Quality of English Language

The manuscript shows several grammatical errors that require correction.

Author Response

Thank you for your comment. We have already calculated the FICI values and added them to the manuscript. Although the FICI provides a single value to summarize the interaction between the agents, the graphs generated by Combenefit show synergistic interactions at different concentration points. These graphs provide a more complete and detailed picture of the interactions, revealing synergies that may not be fully reflected in a single FICI value.

Round 2

Reviewer 2 Report

Comments and Suggestions for Authors

I consider that the manuscript now includes revisions that improved its quality, in addition to the study of the optimization of the combination of essential oils with ampicillin against extended-spectrum beta-lactamase-producing Escherichia coli, gives merit to the work, especially, that the E.coli strains studied were characterized through the Detection of antibiotic resistance genes β-lactamics. However, the antibacterial study was not deep and in the future, for other works, it will be necessary to deepen in mechanisms of action and analysis of other strains of the same species to see if the effect is strain-dependent.

Author Response

The answer is attached

Reviewer 3 Report

Comments and Suggestions for Authors

The authors of the manuscript 3112360 _V2 entitled “ Lippia origanoides and Thymus vulgaris essential oils synergize  with ampicillin against extended-spectrum beta-lactamase-producing Escherichia coli” kept the analysis of the MIC and MBC values and the interaction with the antibiotic in the discussion section what is not according to the author’s instructions of the journal:

 Results: Provide a concise and precise description of the experimental results, their interpretation as well as the experimental conclusions that can be drawn.

Discussion: Authors should discuss the results and how they can be interpreted in perspective of previous studies and of the working hypotheses. The findings and their implications should be discussed in the broadest context possible and limitations of the work highlighted. Future research directions may also be mentioned. This section may be combined with Results.

Moreover, it seems very weird this configuration of presenting a manuscript.

The the Loewe model must be briefly described.

The English language still requires a deep revision.

Comments on the Quality of English Language

The sentence from 209-213 is not clear, the English requires improvement: Resistance genes present in ESBL-producing E. coli strains were identified from plasmid DNA, 65% of the strains were positive for blaTEM, and 15% were positive for blaCTX-M; 210 of the 20 strains, two were amplified for both genes (Figure 2), the positive control, the  resistance genes found were compared with the genes of a previously identified multi-drug-resistant strain.

Line 202 cefotaxime, cefotaxime, cefotaxime

Author Response

The answer is attached
